# Chemotaxis towards autoinducer 2 mediates autoaggregation in *Escherichia coli*

Leanid Laganenka[1], Remy Colin[1] & Victor Sourjik[1]

Bacteria communicate by producing and sensing extracellular signal molecules called autoinducers. Such intercellular signalling, known as quorum sensing, allows bacteria to coordinate and synchronize behavioural responses at high cell densities. Autoinducer 2 (AI-2) is the only known quorum-sensing molecule produced by *Escherichia coli* but its physiological role remains elusive, although it is known to regulate biofilm formation and virulence in other bacterial species. Here we show that chemotaxis towards self-produced AI-2 can mediate collective behaviour—autoaggregation—of *E. coli*. Autoaggregation requires motility and is strongly enhanced by chemotaxis to AI-2 at physiological cell densities. These effects are observed regardless whether cell–cell interactions under particular growth conditions are mediated by the major *E. coli* adhesin (antigen 43) or by curli fibres. Furthermore, AI-2-dependent autoaggregation enhances bacterial stress resistance and promotes biofilm formation.

[1] Max Planck Institute for Terrestrial Microbiology and LOEWE Center for Synthetic Microbiology (SYNMIKRO), Karl-von-Frisch Strasse 16, 35043 Marburg, Germany. Correspondence and requests for materials should be addressed to V.S. (email: victor.sourjik@synmikro.mpi-marburg.mpg.de).

It is well established that under many natural environmental conditions bacteria prefer to exist as multicellular structures, such as surface-attached biofilms or freely floating aggregates. Within these structures cells are protected from various stress factors such as exposure to ultraviolet light, acids, detergents or antimicrobial agents[1,2].

The Gram-negative bacterium *Escherichia coli* is one of the model organisms for studying both cell aggregation and biofilm formation. Autoaggregation in *E. coli* is observed as emergence of microscopic cell clumps that can further lead to macroscopic flocculation and settling of cells in static liquid cultures[3]. A major determinant of autoaggregation in *E. coli* is antigen 43 (Ag43), the abundant outer membrane protein that belongs to the autotransporter family and is secreted via the type V secretion system[4]. During autoaggregation, Ag43 α-subunits of adjacent cells interact in a head-to-tail conformation resulting in dimer formation[5]. While Ag43 is the only known surface factor implicated in autoaggregation of exponentially growing non-pathogenic *E. coli*, pathogenic strains can also aggregate via fimbriae or pili[6–8]. Moreover, at lower growth temperatures (below 30 °C) and in the later growth phase, formation of cellular aggregates can also be mediated by interactions of curli fibres, a major proteinaceous component of *E. coli* biofilm matrix[9–12].

Expression of Ag43 is a classic example of phase variation, where cells in a clonal population can be either in an ON state (expressing Ag43) or in an OFF state. Ag43 phase variation results from binding competition between the repressor OxyR and Dam methyltransferase (methylase) to the regulatory region of *agn43* (alternatively called *flu*). Deletion of *oxyR* leads to a locked-ON state, whereas deletion of *dam* leads to a locked-OFF state[13].

Although bacterial autoaggregation is normally thought of as a passive process, during the mid- to late exponential phase of growth when aggregation becomes apparent *E. coli* cells are highly motile and chemotactic, that is, able to follow gradients of nutrients and other environmental stimuli[14]. These chemotactic stimuli are detected by transmembrane chemoreceptors that regulate activity of the cytoplasmic histidine kinase CheA and subsequent phosphorylation of the response regulator CheY. Phosphorylated CheY binds to flagellar motors and induces a switch from the default counterclockwise to clockwise rotation, promoting cell tumbling. Increased binding of chemoattractants to the sensory domain of receptors—which can be either direct or mediated by periplasmic binding proteins—results in inhibition of the autophosphorylation activity of CheA and decrease in the level of phosphorylated CheY, causing smooth swimming.

Chemotaxis to self-secreted attractants is well known to promote aggregation of eukaryotic organisms, such as social amoebae[15]. It was thus speculated that chemotaxis-dependent aggregation might also exist in bacteria[16], but direct evidence for such behaviour is still missing. In a porous medium or in a microfluidic channel, *E. coli* can indeed form large dynamic clusters where thousands of cells are kept together solely through chemotactic self-attraction[17–21]. However, this behaviour only occurs under specific conditions when *E. coli* secretes high levels of amino acids that act as attractants, and its physiological significance remained unclear. Besides *E. coli*, regulation of motility by the chemotaxis pathway has been shown to affect autoaggregation (clumping) of *Azospirillum brasiliense*, but this regulation is inhibitory and chemotaxis itself is not required for aggregation[22].

In this study, we provide evidence for the involvement of motility and chemotaxis in the Ag43-dependent aggregation of *E. coli*. Our results suggest that aggregation should be seen as an active process that requires not only specific adhesins but also swimming to promote random intercellular collisions and subsequent chemotactic response to gradients of self-produced attractant. We further show that this self-attraction is mediated by the quorum-sensing molecule autoinducer-2 (AI-2), the only quorum-sensing signal described for *E. coli*. Chemotaxis to AI-2 similarly promotes aggregative behaviour mediated by curli fibres in cells grown at low temperature to the early stationary phase. Our results demonstrate that such chemotaxis-driven aggregation enhances AI-2-mediated signalling, biofilm formation and stress resistance.

## Results

**Aggregation of *E. coli* depends on motility and chemotaxis.** We first investigated aggregation of *E. coli* strain W3110 (RpoS⁺)[12] grown at 37 °C to a mid-exponential growth phase. Consistent with previous studies performed for other *E. coli* K-12 strains[8], under these conditions the high-density ($OD_{600}$ of 2.0) culture of W3110 showed reproducible aggregation that was dependent on Ag43 (Supplementary Fig. 1) but not on curli or on other biofilm matrix components, poly-beta-1,6-N-acetyl-D-glucosamine and colanic acid (Supplementary Fig. 2). This result confirms that a large fraction of W3110 cells is in the ON state of *agn43* expression under our conditions. Nevertheless, to avoid potential complexity associated with the phase variation of *agn43* expression between and within individual cultures[23], we subsequently used overexpression of the Dam methyltransferase to lock our strains in the ON state (Supplementary Fig. 3). The Dam⁺⁺ wild-type strain aggregated even more efficiently than the original W3110 wild type (compare Fig. 1a and Supplementary Fig. 1). Aggregation was also dependent on density of the cell culture, being negligible below $OD_{600}$ of 0.25, relatively constant in the $OD_{600}$ range between 0.5 and 2.0, and further increasing at $OD_{600}$ of 6.0 (Fig. 1a and Supplementary Fig. 4).

Notably, we observed that aggregation was completely abolished not only by deletion of *agn43* (Δ*flu*) (Fig. 1b) but also on deletion of *fliC* that encodes flagellin (Fig. 1c and Supplementary Fig. 1). A similar defect was observed for a Δ*motA* mutant that lacks a major component of the flagellar motor and has paralysed flagella (Fig. 1d). Aggregation of Δ*flu*, Δ*fliC* and Δ*motA* strains could not be restored even at very high cell density, meaning that both Ag43 and flagella-driven motility are strictly required for this process.

Furthermore, we observed that aggregation was severely affected in the non-chemotactic (but motile) Δ*cheY* mutant (Fig. 1e and Supplementary Fig. 1). In contrast to motility-deficient strains Δ*cheY* cells still formed small aggregates. However, at $OD_{600} = 1.0$ these aggregates did not reach the size of the wild-type structures (Fig. 1f). This requirement of chemotaxis for aggregation was even more pronounced at lower cell densities, but it was largely alleviated at very high cell density when aggregation of Δ*cheY* and wild-type cells became comparable (Fig. 1e and Supplementary Fig. 4). We thus concluded that while chemotaxis (as opposed to motility) is not absolutely essential for aggregation, it strongly enhances aggregation at lower cell densities, likely by mediating attraction of individual motile cells towards aggregates. We further observed that aggregation requires chemoreceptor Tsr, which is highly abundant in exponentially growing *E. coli* cells[24], whereas the deletion of Tar, another major chemoreceptor, had a milder effect (Fig. 1f and Supplementary Fig. 1).

**Chemotaxis to AI-2 enhances aggregation.** The involvement of Tsr in aggregation was particularly interesting, since this receptor has been previously shown to mediate chemotactic response to AI-2, a universal quorum-sensing molecule that can be produced

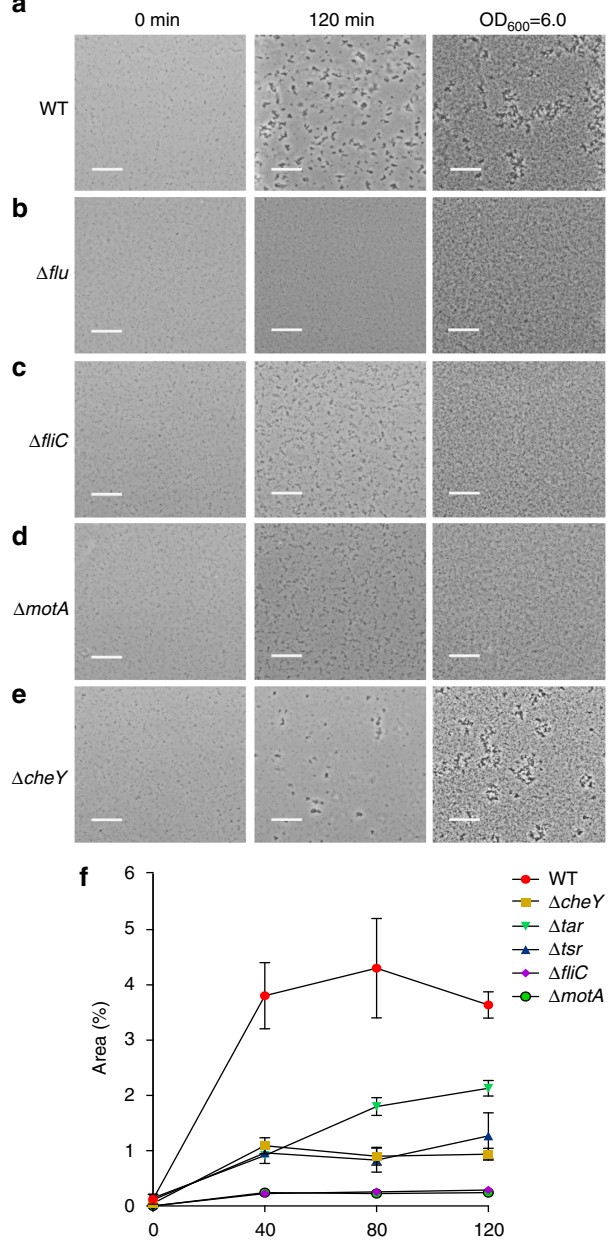

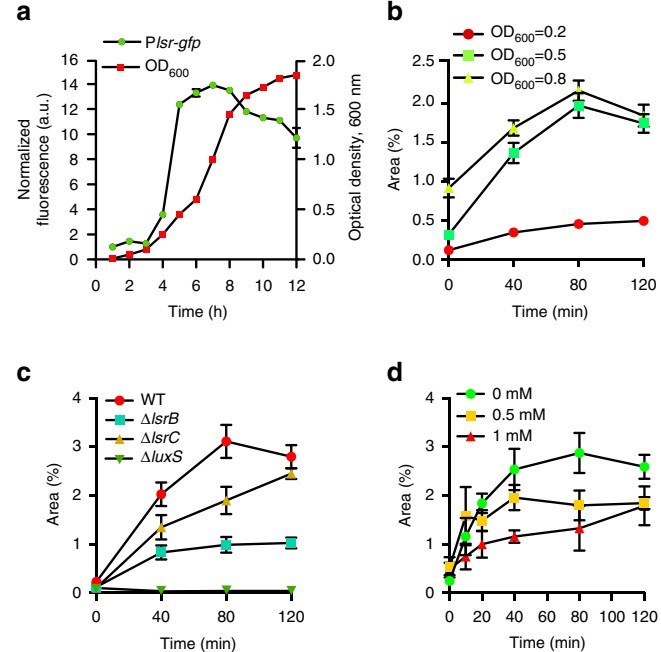

**Figure 2 | Autoaggregation depends on production and sensing of autoinducer 2.** (**a**) Activity of the *lsr* promoter reporting AI-2 levels during growth of the wild-type population as a function of the growth stage of *E. coli* culture. (**b**) Aggregation of the wild type at different stages of growth, assayed as in Fig. 1. (**c**) Aggregation of strains defective in production (Δ*luxS*), sensing and import (Δ*lsrB*) or only import (Δ*lsrC*) of AI-2. (**d**) Effects of indicated concentrations of added DPD on aggregation of the wild-type cells. Error bars in all panels indicate s.d. of three independent replicates.

**Figure 1 | Swimming motility and chemotaxis are required for autoaggregation.** (**a**–**e**) Aggregation of the wild-type *E. coli* W3110 (**a**), and its Δ*flu* (**b**), Δ*fliC* (**c**), Δ*motA* (**d**) and Δ*cheY* (**e**) knockouts grown to $OD_{600}$ of 0.6 at 37 °C. For aggregation experiments, cells were resuspended in fresh TB medium to $OD_{600}$ of 1.0 (two left panels) or 6.0 (right panel), and aggregation was assayed at room temperature in channels using microscopy as described in Methods. Aggregation at $OD_{600}$ of 6.0 is shown at 120 min. Scale bars, 20 μm. (**f**) Quantification of the area occupied by aggregates in microscopy images for the wild type and indicated mutants. Error bars indicate s.d. of three independent replicates.

and recognized by many bacteria[25,26]. AI-2 is a by-product of the activated methyl cycle, required for recycling of *S*-adenosyl-L-methionine. In *E. coli*, AI-2 is initially produced by the enzyme LuxS as (*S*)-4,5-dihydroxy-2,3-pentandione (DPD) and later undergoes spontaneous cyclization. *E. coli* possesses a specific ATP-binding cassette transporter for AI-2 uptake, which includes the periplasmic binding protein LsrB, the

cytoplasmic ATP-binding protein LsrA, as well as two membrane proteins, LsrC and LsrD, that form the transport channel[25]. The imported AI-2 undergoes phosphorylation by LsrK and then binds to and activates LsrR, which leads to de-repression of the *lsr* operon and possibly other genes[25]. Besides its function in transport, AI-2-bound LsrB was proposed to interact with the sensory domain of Tsr, thereby eliciting a chemotactic response[27].

*E. coli* is known to primarily produce and secrete AI-2 during the mid- to late exponential growth phase[25] (Fig. 2a), which is seemingly consistent with the growth phase dependence of autoaggregation (Fig. 2b). Together with the requirement of Tsr for aggregation, this indicated that AI-2 might be the aggregation-promoting chemotactic signal. This hypothesis was supported by strongly reduced aggregation of Δ*luxS* and Δ*lsrB* mutants that are impaired in production and perception of AI-2, respectively (Fig. 2c, Supplementary Figs 1 and 5 and Supplementary Fig. 5). Although the deletion of *luxS* also resulted in decreased motility (Supplementary Fig. 5a) thus complicating data interpretation, Δ*lsrB* showed normal motility and chemotaxis towards amino acids while its aggregation was reduced to the level of Δ*cheY* strain. This defect of Δ*lsrB* strain in aggregation was apparently due to its deficiency in the sensing of AI-2 and not because of the deficient uptake, as deletion of *lsrC* that similarly abolishes the AI-2 uptake led to only a minor reduction of aggregation (Fig. 2c). Consistent with the involvement of AI-2 signalling, addition of synthetic DPD/AI-2 (that is, DPD that spontaneously converted to AI-2) to the wild-type culture reduced aggregation in a dose-dependent manner (Fig. 2d).

To directly confirm that AI-2 is a specific LsrB-dependent chemoattractant for *E. coli*[27], we studied *E. coli* behaviour in

microfluidic gradients of the synthetic DPD/AI-2, using a non-metabolizable attractant α-D,L-methylaspartic acid (MeAsp) as a positive control (Supplementary Fig. 6a,b). The movement of wild-type cells showed a pronounced chemotactic bias up the 0–10 μM gradient of DPD/AI-2 (Supplementary Fig. 6c), comparable to the bias observed for the 0–200 μM gradient of MeAsp. As expected, Δ*lsrB* strain showed no chemotaxis to AI-2. Notably, wild-type cells were no longer able to follow the same gradient of DPD/AI-2 above the background of 200 μM. This is consistent with a narrow dynamic range of attractant concentrations that can be sensed via a periplasmic binding protein[28], due to saturation of the sensor at high background stimulation. Importantly, no apparent effect of even high background concentration of AI-2 on swimming speed or chemotaxis to MeAsp was observed (Supplementary Fig. 6c,d). Taken together, these results strongly suggest that AI-2 acts as a specific chemoattractant during autoaggregation of *E. coli*.

**Chemotaxis governs kinetics of aggregation and disaggregation**. To better understand the dynamics of autoaggregation and its dependence on AI-2 chemotaxis, we followed its early stage with high time resolution and at lower cell density (OD$_{600}$ = 0.5) (Fig. 3a–c, and Supplementary Movies 1 and 2). We observed that the aggregation of the wild-type cells proceeds very rapidly, with ∼ 50% of cells being incorporated into the aggregates already during the first 5 min of observation (∼ 8 min after the cells were loaded into the channel; Fig. 3a,b). During this phase, both the number (Fig. 3b) and size of aggregates (Fig. 3c) increased. Subsequently, the aggregation slowed down, reaching a peak at about 1 h, when the majority of cells became incorporated into the aggregates (Supplementary Movie 3). Nevertheless, individual aggregates remained highly dynamic in their size (Supplementary Fig. 7, and Supplementary Movies 1 and 3). Such dynamics demonstrates reversibility of cell incorporation into Ag43-mediated aggregates, which is consistent with a relatively low strength of interactions mediated by Ag43 (ref. 5).

The rate of aggregation was significantly lower for Δ*cheY* and Δ*lsrB* strains (Fig. 3a–c and Supplementary Movie 2), with both the number and the size of aggregates increasing slower than in the wild type. Notably, this difference in the aggregate size and number was already observed at the early stage of aggregation, when the size of aggregates was below 50 μm. The rate of aggregation and particularly the number of aggregates were also moderately decreased in Δ*lsrC* cells that are deficient in AI-2 uptake, indicating that degradation might play a role in sharpening the AI-2 gradients.

Consistent with the reversible nature of cell association within aggregates, we further observed that after 2 h of incubation, aggregates formed by the wild-type cells began to disperse again (Fig. 3d), despite individual cells remaining highly motile (Supplementary Movie 4). Dispersal was not observed for Δ*cheY* or Δ*lsrB* aggregates, which remained at approximately constant size for over 5 h, suggesting that it might be caused by the loss of chemotaxis towards aggregates. We hypothesized that this loss of chemotaxis is likely to be explained by excessive accumulation of AI-2 in the medium during the experiment, which disables sensing of AI-2 gradients above this high background (Supplementary Fig. 6). Indeed, quantification of AI-2 in the cell-culture supernatant using transcriptional reporter showed that levels of AI-2 continued to rise at the onset of disaggregation (Fig. 3e and Supplementary Fig. 8). Consistent with an excess rather than depletion of AI-2 being a cause of disaggregation, similar dispersal kinetics was observed for the uptake-deficient Δ*lsrC* and wild-type cells (Fig. 3d).

Further supporting our hypothesis, addition of external synthetic DPD/AI-2 (0.2 mM) even at later stages of autoaggregation greatly reduced the growth of the aggregates (Supplementary Fig. 9a,d), although the effect is presumably weakened by rapid consumption of AI-2. Similar inhibition and even moderate dispersal of the aggregates could be achieved by very high levels (10 mM) of Tsr ligand L-serine, which are known to generally disable chemotaxis[29]. Even more pronounced dispersal effect was observed on incubation with 2-aminoisobutyric acid (AIbu), a less chemoattractive but non-metabolizable analogue of L-serine[30], consistent with our assumption that degradation weakens the dispersal effects that are induced by added chemoattractants. Importantly, no dispersal was observed for Δ*cheY* and Δ*lsrB* aggregates on addition of AI-2 or other chemoattractants (Supplementary Fig. 9b,c,e,f), which confirms the key role of the chemotaxis inhibition in the dispersal process.

**Aggregation promotes AI-2 signalling**. We next explored the consequences of such chemotaxis-mediated autoaggregation on the AI-2-dependent transcriptional response. We speculated that high density of AI-2-secreting cells within aggregates might lead to more efficient response induction. Indeed, when cultures were incubated under conditions that enable aggregation, the induction of the P$_{lsr}$-*egfp* reporter in the wild type—but not in the aggregation-deficient Δ*flu*, Δ*cheY* or Δ*lsrB* mutants—increased significantly during the first 40 min (Fig. 4a). This was likely explained by the increased local accumulation of AI-2 in the aggregates, because the wild-type strain overexpressing LuxS showed a steadily high level of *lsr* operon expression.

Consistent with the proposed higher induction of the reporter in only a fraction of cells within aggregates, wild-type cells showed a broader distribution of the levels of reporter fluorescence than aggregation-deficient strains or than the wild-type overexpressing LuxS (Fig. 4b). To further confirm this interpretation, reporter fluorescence was imaged within aggregates and in non-aggregated planktonic cells using confocal microscopy. Aggregated cells had indeed higher level of *lsr* operon induction (Fig. 4c), in contrast to the control cell expressing only *egfp*.

**Aggregation enhances stress resistance and biofilm formation**. Aggregation is known to enhance stress resistance of bacteria[2], including *E. coli*[31]. Consistent with these previous observations, wild-type cells that were treated with H$_2$O$_2$ had much higher rate of survival under conditions that favour aggregation (Fig. 5a). In contrast, no increase in survival was observed in non-aggregating Δ*flu* cells, even on overexpression of LuxS. On the basis of these results, we conclude that the increased oxidative stress resistance is indeed mediated by physical protection of cells within aggregates rather than by the quorum-sensing signalling.

Although Ag43 is usually not considered as a major determinant of biofilm formation, the effect of *flu* deletion on surface-attached biofilms has been reported[32]. We thus tested possible involvement of the Ag43- and chemotaxis-dependent aggregation in biofilm formation. Indeed, crystal violet staining of surface-attached biofilms grown for 24 h at 37 °C in microtitre plates revealed significant decrease in biofilm formation in Δ*flu*, Δ*cheY* and Δ*lsrB* strains (Fig. 5b). Microscopic observation of biofilms formed under such static conditions demonstrated that biofilms formed by Δ*lsrB*, Δ*cheY* or Δ*flu* cells were less structured than the wild-type biofilm (Fig. 5c). This was further confirmed by quantification of the microcolony volumes in biofilm images, which showed that the wild type formed significantly larger microcolonies than Δ*cheY* or Δ*lsrB*, and Δ*flu* formed no

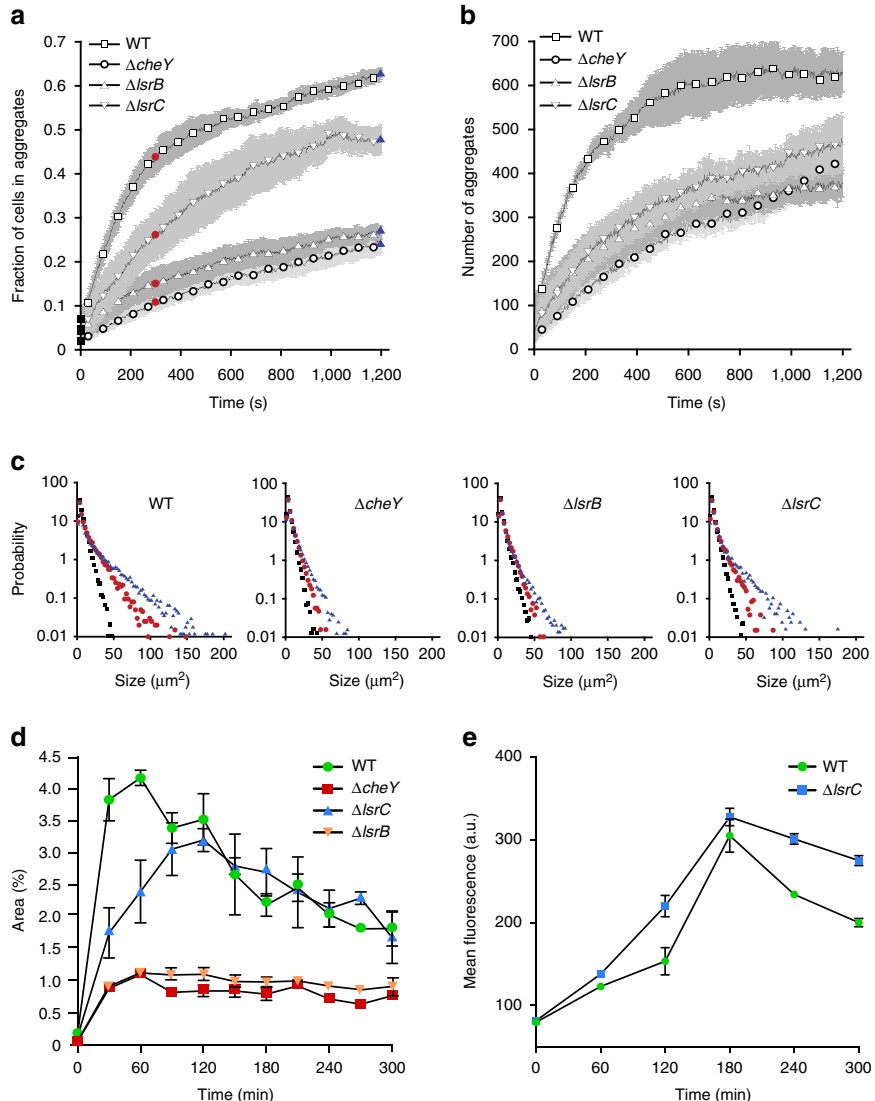

**Figure 3 | Kinetics of initial aggregation and dispersal depend on AI-2.** (**a–c**) Time dependence of the fraction of cells in aggregates (**a**), number of aggregates (**b**) and aggregate size (**c**). Distribution of aggregate sizes in **c** is shown at 0 (black squares), 5 (red circles) and 20 min (blue triangles), as indicated in **a**. Aggregation was assayed as in Fig. 1, except $OD_{600}$ was adjusted to 0.5. (**d,e**) Dynamics of aggregation and dispersal for the wild-type, $\Delta cheY$, $\Delta lsrB$ and $\Delta lsrC$ strains over 5 h incubation (**d**) and corresponding levels of AI-2 activity in supernatants quantified using fluorescence reporter strain as described in Methods. Reporter fluorescence was measured using flow cytometry and expressed in arbitrary units (a.u.) of fluorescence. Aggregation was assayed as in Fig. 1. Error bars in all panels indicate s.d. of three independent replicates.

detectable microcolonies (Supplementary Fig. 10). Consistent with its greater importance for aggregation, deletion of Ag43 had a more pronounced effect on biofilm formation than the lack of AI-2 chemotaxis.

**AI-2 chemotaxis enhances curli-mediated aggregation.** Although curli fibres are not important for autoaggregation of *E. coli* cells grown to late exponential phase at 37 °C, curli expression is highly upregulated at lower temperatures (below 30 °C) and higher optical density (OD). Consistent with previous reports[10,11], we observed that aggregation behaviour of cell grown at 30 °C to $OD_{600} = 1.0$ was dependent on curli, whereas *flu* deletion had only a moderate effect (Fig. 6a and Supplementary Fig. 11a). Despite this different mode of cell–cell interaction and lower expression of *lsr* operon at 30 °C (Supplementary Fig. 12a), the dependence of curli-mediated aggregation on motility and AI-2 chemotaxis was similar to the Ag43-mediated aggregation. However, aggregates formed under

these conditions showed little dispersal even after 3.5 h of observation (Supplementary Fig. 11b), presumably due to the lower rate of AI-2 accumulation at 30 °C and/or higher stability of curli-mediated interactions. Consistently, biofilm formation at 30 °C was affected by *cheY* and *lsrB* deletions, whereas *flu* deletion had lesser effect in this case (Fig. 6b), as confirmed by the crystal violet staining (Supplementary Fig. 12b) and quantification of the microcolony size distribution within biofilms (Supplementary Fig. 12c).

## Discussion
The ability to form multicellular agglomerations, either suspended aggregates or surface-attached biofilms, is widespread among bacteria. Multicellular structures provide a number of benefits, such as increased resistance to various stress factors that are essential for bacterial survival under changing environmental conditions. Despite the importance of bacterial aggregation in the

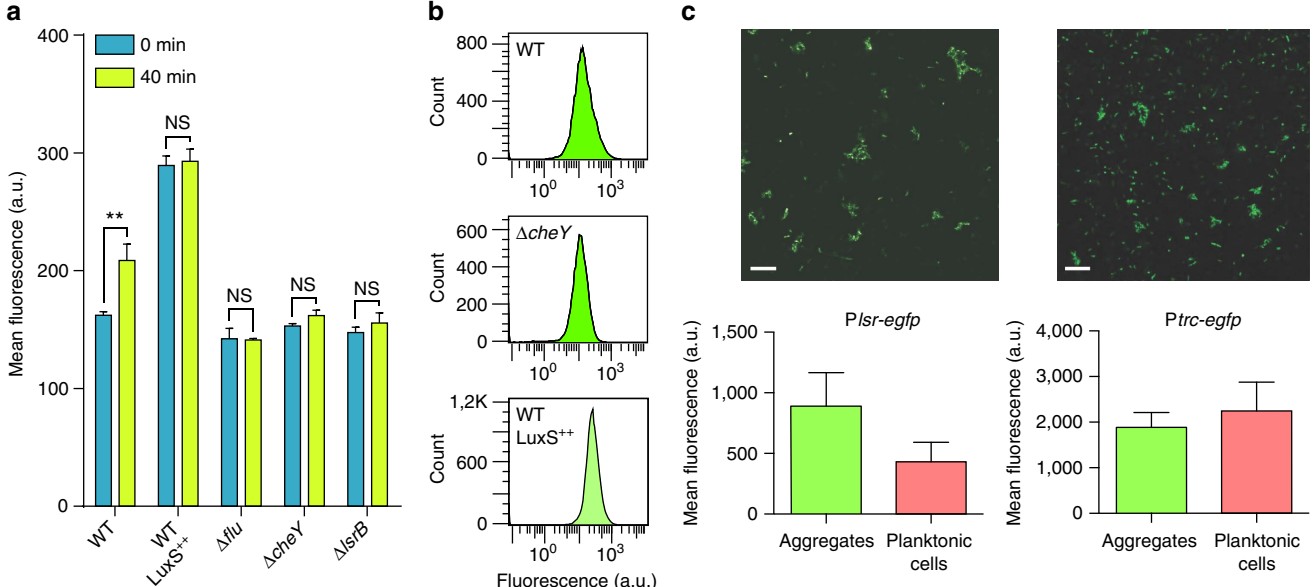

**Figure 4 | High cell density in aggregates promotes induction of AI-2 response.** (**a**) *lsr* promoter activation in the aggregating wild-type and non-aggregating Δ*cheY*, Δ*flu* and Δ*lsrB* strains, as well as in the wild-type overexpressing LuxS. *P* values were calculated using Mann–Whitney test (**$P < 0.05$, NS, not significant, $P > 0.05$). (**b**) Distribution of fluorescence levels of P$_{lsr}$-*egfp* at 40 min of aggregation in the wild type, Δ*cheY* (Mann–Whitney test, $P < 0.05$, $n = 3$) and in the wild-type overexpressing LuxS (Mann–Whitney test, $P < 0.05$, $n = 3$), measured using flow cytometry and expressed in arbitrary units (a.u.) of fluorescence. (**c**) Confocal microscopy images and corresponding quantification of P$_{lsr}$-*egfp* fluorescence in aggregates and in planktonic cells of the wild-type population (left, Mann–Whitney test, $P < 0.0001$) and of cells expressing *egfp* from the IPTG-inducible *trc* promoter (right, Mann–Whitney test, $P > 0.05$). Scale bars, 20 μm. Error bars in all panels indicate s.d. of three independent replicates.

environmental context, current understanding of this behaviour is still limited, even for such model organism as *E. coli*.

Flagellar motility and chemotaxis are normally associated with the behaviour of individual planktonic cells, and formation of biofilms is viewed as an irreversible transition from motile to the sessile lifestyle[1,33,34]. Although under some conditions flagella and motility are known to contribute to formation of surface-associated biofilms[35,36], these effects have been interpreted in the context of bacteria–surface interactions and not of cell–cell interactions. It was proposed that flagella may function as surface adhesins[37] or sensors[38,39], or that swimming may promote transient cell-surface contacts[40]. The role of chemotaxis in biofilm formation has not been demonstrated to date, and *E. coli* chemotaxis mutants can form normal biofilms under standard conditions[32,35].

Even less explored are the functions of motility and chemotaxis in formation of suspended cell aggregates. In *E. coli*, autoaggregation was proposed to be mutually exclusive with motility[41]. In *A. brasilense*, aggregation (clumping) is affected by the deletion of Che1 chemotaxis pathway that modulates swimming velocity[42], suggesting an important role for motility. However, the Che1-mediated tactic response itself is not required for aggregation since deletion of the pathway increases rather than decreases clumping[22].

Our study thus provides the first direct evidence that chemotaxis towards a self-secreted attractant mediates autoaggregation of bacteria. Notably, motility and chemotaxis are required regardless of the adhesin that mediates cell interactions in *E. coli*—Ag43 during exponential growth at 37 °C or curli during early stationary phase at 30 °C. According to our observations, the aggregation process could be principally divided into three phases (Fig. 7). Phase I corresponds to the initial formation of 'seeding' aggregates by random collisions of motile cells, which does not require chemotaxis. Consistent with that, formation of small aggregates is observed for Δ*cheY* but not for

Δ*fliC* or Δ*motA* cells (Fig. 1). At physiological cell densities, however, further growth of these aggregates through purely random collisions is relatively inefficient. Instead, during the next phase (phase II) these seeding aggregates apparently secrete sufficiently high levels of AI-2 to mediate gradient formation and chemotactic attraction of individual cells to the aggregates. Such chemoattraction greatly enhances the rate of aggregate growth, likely by increasing local cell density. Consistent with that, chemotaxis has a more pronounced effect on aggregation at lower densities of the cell culture (Fig. 1a,e and Supplementary Fig. 4). Chemoattraction results in a transient phase of rapid aggregate growth that subsequently slows down as the aggregates reach equilibrium with free-swimming cells, which is strongly shifted towards aggregates in the wild-type compared with non-chemotactic cells (Fig. 3). Subsequent gradual increase in the background levels of AI-2 in the medium disrupts chemotaxis towards aggregates, because cells are no longer able to follow gradients of AI-2 above high background, thus inducing aggregate dispersal (phase III). Supporting this explanation, during dispersal the levels of aggregation in the wild-type cells gradually approached those in the non-chemotactic or AI-2-insensitive strains, whereas these latter strains showed no significant dispersal. Dispersal phase is particularly prominent for the Ag43-mediated aggregation, whereas curli-mediated interactions seem to be significantly more stable.

Aggregation mediated by the AI-2 taxis appears to have several physiological consequences. First, we observed that high local cell density within aggregates promotes AI-2 signalling in comparison with non-aggregating cells. *E. coli* can thus use autoaggregation to reach critical densities for quorum sensing already at low overall density of the population. Such local induction of quorum-sensing response within aggregates is consistent with a theoretical concept of efficiency sensing[43], as well as with a previously observed local quorum-sensing response induction in *Vibrio harveyi*[21].

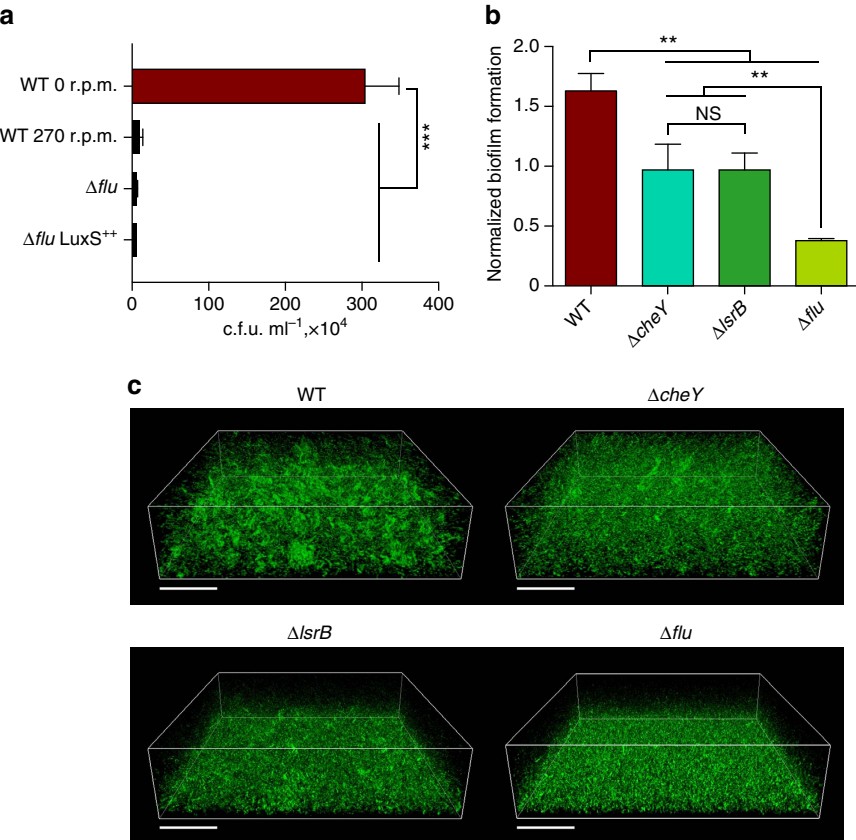

**Figure 5 | Ag43-dependent aggregation promotes resistance to oxidative stress and biofilm formation.** (**a**) Aggregation improves viability under oxidative stress conditions. Survival of the wild-type and $\Delta flu$ cells on exposure to 0.5% $H_2O_2$ under aggregation (no shaking) or non-aggregation (shaking at 270 r.p.m.) conditions. Overexpression of LuxS does not rescue viability defect on the autoaggregation-defective $\Delta flu$ strain. P value was calculated using Mann–Whitney test (***$P < 0.01$). (**b**) Biofilm formation under static culture grown at 37 °C for 24 h, quantified using crystal violet (CV) staining. Shown in arbitrary units (a.u.) are CV values normalized by the optical density. P values were calculated using Mann–Whitney test (**$P < 0.05$, NS, not significant, $P > 0.05$). Error bars in **a,b** indicate s.d. of five independent replicates. (**c**) Confocal laser scanning microscopy of biofilms formed by the wild-type strain and by the aggregation-deficient mutants after 24 h static culture grown at 37 °C. Scale bars, 40 µm.

Although AI-2 is produced by many bacteria and is used for communication both within and between species, there are few characterized examples of physiological functions of AI-2 signalling[25]. Consequently, the importance of *E. coli* response to AI-2 beyond its uptake and metabolism remained unclear to date, although metabolism of AI-2 might itself provide bacteria with a benefit[44] and/or with a competitive advantage in mixed communities[45]. Building on a previous observation[27] of the chemotactic response to AI-2, here we characterized the first uptake-independent function of AI-2 signalling in *E. coli*. Importantly, AI-2-mediated autoaggregation not only promotes local AI-2 signalling but it also provides physical protection against oxidative stress and contributes to the development of surface-attached biofilms. Our results thus provide evidence for the long-hypothesized role of self-attraction in cell aggregation and in biofilm formation. Given the ubiquitous nature of AI-2 production by bacteria, it is likely that chemotaxis to AI-2 is involved in aggregation, and possibly even co-aggregation, of other species.

## Methods

**Bacterial strains and growth conditions.** The strains and plasmids used in this study are listed in Supplementary Table 1. All strains were derived from *E. coli* W3110 (RpoS⁺)[12]. Cells were grown either on 1.5% Luria Bertani (LB) agar or in liquid tryptone broth (TB) medium (10 g tryptone and 5 g NaCl per litre) supplemented with antibiotics, where necessary. Gene deletions were obtained via

PCR-based inactivation of chromosomal genes[46] or using P1 transduction[47]. Km^R cassettes were eliminated via FLP recombination[48].

To lock the ON state of Ag43 production, cells were transformed with a high-copy number plasmid (pVS1722) encoding Dam methyltransferase under control of a *trc* promoter inducible by isopropyl β-ᴅ-1-thiogalactopyranoside (IPTG). However, no induction was used, since the basal expression of Dam from pVS1722 was sufficient to abolish the Ag43 phase variation.

**Autoaggregation assay.** *E. coli* cells were grown overnight in TB with appropriate antibiotics, diluted 1:1,000 and grown at 37 °C with shaking at 200 r.p.m. to $OD_{600}$ of 0.5–0.6, unless stated otherwise. Where indicated, cells were alternatively grown at 30 °C to $OD_{600}$ of 1.0. Cells were then collected by centrifugation (5 min, 4,700 r.p.m.) and resuspended in TB to final $OD_{600}$ of 1.0 (or 2.0 for cells with the native level of *dam* expression), unless stated otherwise. Cell suspensions were loaded into ibidi channels (µ-Slide Chemotaxis³ᴰ; ibidi GmbH, Germany) and cell clumping was observed at room temperature (20 °C) using phase-contrast microscopy (Nikon TI Eclipse, ×10 objective, numerical aperture = 0.3, CMOS camera EoSens 4CXP). Images were analysed using Particles Analysis Tool (ImageJ, http://imagej.nih.gov/ij/) to determine the area occupied by aggregates. Where indicated, synthetic DPD solution (provided by Dr Rita Ventura, ITQB, Oeiras, Portugal)[49] was added to cell suspensions (because DPD spontaneously converts to AI-2, we referred to it as DPD/AI-2).

Aggregation kinetics was recorded at the final $OD_{600} = 0.5$ and at a frame rate of 0.5 frames per second. Image in each frame was corrected for the uneven illumination in the large field of view (1.2 × 0.72 mm), using the Fit Polynomial filter of ImageJ to subtract the background approximated as a fourth-order polynomial. *E coli* cells and their aggregates were identified using a threshold (−10 grey levels) relative to the evened background and analysed for their size using a custom-written particle-tracking algorithm. Aggregates were defined as objects with the size above 50 px², with single cells being on average ~10 px². The exact value of the size threshold for aggregate assignment did not qualitatively affect the results.

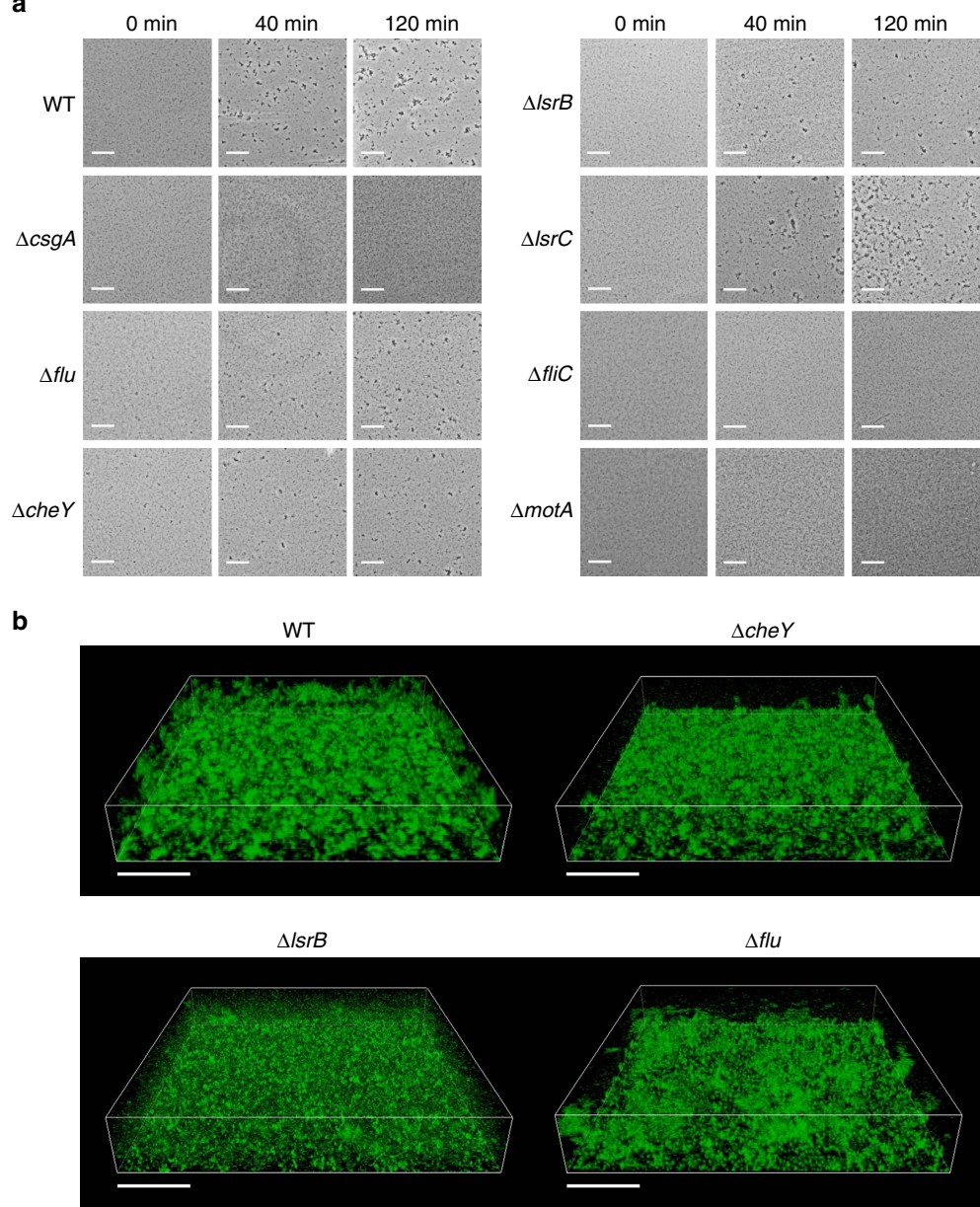

**Figure 6 | Chemotaxis to AI-2 enhances curli-mediated aggregation and biofilm formation. (a)** Aggregation of the wild-type *E. coli* W3110 cells and indicated knockout mutants grown to $OD_{600}$ of 1.0 at 30 °C, and assayed as in Fig. 1 but at $OD_{600}$ of 2.0. **(b)** Confocal laser scanning microscopy of biofilms formed at 30 °C after 24 h static culture growth. In both experiments, cells with native level of Dam expression were used. Scale bars, 40 μm.

**Immunodetection of Ag43**. For the immunoblot analysis of Ag43 production[23], cells were grown at 37 °C to $OD_{600} = 0.6$ as described above, collected by centrifugation at 5,000 r.p.m. for 5 min and adjusted to the final $OD_{600}$ of 1.0. Samples (100 μl) were separated using 10% SDS–polyacrylamide gel electrophoresis and transferred to the nitrocellulose membrane using western blotting. A polyclonal rabbit antiserum raised against the α-domain of Ag43 (a gift of Dr Christophe Beloin, Institute Pasteur, Paris, France) was used at a dilution 1:10,000 for immunodetection of Ag43.

**Hydrogen peroxide treatment**. Cell survival on treatment with $H_2O_2$ was tested as described previously[31]. Briefly, cells were grown to $OD_{600}$ of 0.6 as described above, washed once and resuspended in TB at final $OD_{600}$ of 1.0, and 100 μl cell aliquots containing ∼$10^9$ colony-forming units per ml were incubated for 1 h at room temperature in a microtitre plate. Subsequently 100 μl of 1% $H_2O_2$ was added to each sample, incubated for 15 min at room temperature, and cells were washed in TB and plated at appropriate dilutions to determine the number of surviving cells.

**Flow cytometry**. Activity of the *lsr* promoter was assayed using a plasmid-based *egfp* reporter that contains the 217 nucleotide region upstream of the *lsrA* gene.

Samples for flow cytometry were prepared as described above, diluted 1:20 in tethering buffer (10 mM $KH_2PO_4$, 100 μM EDTA, 1 μM L-methionine and 10 mM lactic acid, pH = 7.0) and fluorescence was measured with BD LSRFortessa SORP cell analyser (BD Biosciences, Germany). Before the measurements, cell aggregates were dispersed by vigorous mixing.

The same reporter transformed in Δ*luxS* strain was used as a biosensor to quantify levels of AI-2 in supernatants. Cell-free supernatants were prepared by filtration of liquid cultures through 0.2 μm filter, and 20 μl aliquots of the reporter strain ($OD_{600} = 0.5$) were added to each sample followed by 40 min incubation at 37 °C. The reporter was calibrated using defined concentrations of synthetic DPD/AI-2.

**Biofilm formation**. Biofilm formation in polystyrene microtitre plates was tested using crystal violet staining assay[50]. Briefly, overnight cultures of samples were diluted in TB to $OD_{600}$ 0.05, and 300 μl of each sample was added into the wells of 96-well plate (Corning Costar, flat bottom; Sigma-Aldrich, Germany). After 24 h of incubation at 37 °C, the $OD_{600}$ of the samples was measured, the wells were rinsed with $H_2O$ and 300 μl of 1% crystal violet solution was added to each well. After 15 min incubation at room temperature, the wells were rinsed three times with $H_2O$. Remaining crystal violet was solubilized by adding 300 μl of 96% ethanol, and

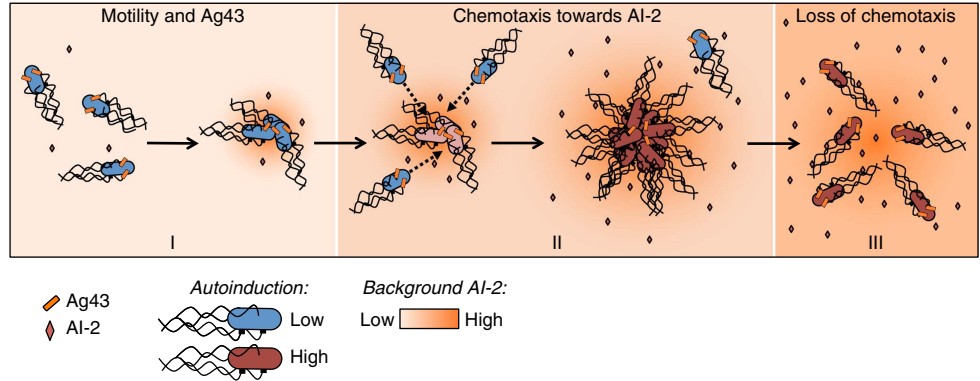

**Figure 7 | Proposed model of autoaggregation.** Ag43-mediated autoaggregation of *E. coli* can be schematically divided into three phases. Phase I corresponds to formation of initial 'seeding' aggregates by random collisions of Ag43-expressing motile cells. These initial aggregates subsequently grow during phase II. Here gradients of AI-2 that are produced by the aggregates serve to attract additional cells. Higher local cell density in the aggregates promotes AI-2-mediated autoinduction (de-repression of the *lsr* operon). Owing to relative weakness of Ag43-mediated interactions, a fraction of individual swimming cells remains at an equilibrium with the aggregates, and cells continue leaving and joining aggregates during this phase. Finally, phase III reflects dispersal of the aggregates. During this phase, increased levels of AI-2 in the culture lead to the loss of chemotaxis due to high background. This results in the subsequent gradual disaggregation, because more cells detach from aggregates than join them. Phases I and II are similar for curli-mediated aggregation, whereas phase III is less pronounced because of greater stability of interactions.

the $OD_{595}$ of the solution was measured. Values of crystal violet staining were normalized for each sample by the respective $OD_{600}$.

For biofilm imaging, overnight cultures carrying a high-copy number plasmid pVS1515 encoding *egfp* were diluted in TB to $OD_{600}$ 0.05 and grown with shaking at 30 °C or 37 °C to the mid-exponential phase ($OD_{600} = 0.6$). The samples were then once again diluted in fresh TB containing 5 μM IPTG to $OD_{600} = 0.05$, and 350 μl of each sample was loaded into the wells of 8-well glass bottom slides (μ-Slide, 8-well glass bottom; ibidi). The cultures were grown at 30 and 37 °C for 24 h without shaking.

**Confocal laser scanning microscopy.** The biofilm formation was visualized using Zeiss LSM-800 microscope equipped with Apochromat × 40 objective. Z-stack image processing and analysis were performed using ZEN Black software (Zeiss). Quantification of microcolonies in mature biofilms was performed using 3D Objects Counter plugin for ImageJ[51], with Image segmentation threshold set to 30.

Same set-up was used to quantify levels of $P_{lsr}$-*egfp* expression in cell aggregates, prepared as described before. Images were analysed with ImageJ to evaluate the fluorescence intensity of the aggregates as well as of individual cells.

**Chemotaxis assays.** Chemotaxis assays were performed in microfluidic chemotaxis chambers[52,53] that consisted of two reservoirs linked via a small channel with the length of 2 mm and width of 1 mm (Supplementary Fig. 6a), which was imprinted in a poly-di-methylsiloxane layer covalently bound to a microscopy glass slide. Cells were grown as described previously, washed thrice with motility buffer (MB; 10 mM $KPO_4$, 0.1 mM EDTA and 67 mM NaCl, pH 7), and resuspended to a final $OD_{600} = 1$ in MB supplemented with 0.5% glucose (MBg). Cells were stored for 20 min in the fridge to reduce metabolic activity.

To measure chemotaxis towards synthetic DPD/AI-2, one of the reservoirs was filled with a cell suspension and the other with a MBg solution containing either 0 (negative control) or 10 μM DPD/AI-2. Where indicated, further 200 μM DPD/AI-2 was added to both reservoirs. No cells were added to the high-concentration reservoir, because DPD/AI-2 consumption by the cells could abolish the gradient. To test chemotaxis towards MeAsp in the presence of DPD/AI-2 (Supplementary Fig. 6a), the reservoirs were filled with cell suspensions containing 200 μM background DPD/AI-2 and either 0 or 200 μM of MeAsp. Since *E. coli* does not metabolize MeAsp, cells were added on both sides. As demonstrated previously[52], diffusion of chemoattractants from the high-concentration reservoir through the channel creates a linear gradient of concentration within 1 h.

The motion of the bacteria was observed using phase-contrast microscopy (magnification × 10, numerical aperture = 0.3) in the middle of the channel and recorded using a CMOS camera (Eosens 4CXP) at 100 frames per second. The average swimming speed ($v_0$) and the fraction of cells swimming (α) were determined using differential dynamic microscopy[52]. The average net drift of the population of cells $v_{drift}$ was measured using phase differential microscopy[54]. The chemotactic bias was defined as $b = v_{drift}/αv_0$, which estimates the velocity-independent movement of the population of swimming cells in the gradient. In the case of diffusion alone, this bias can be estimated as $b_{diff} = l/(3(1 - cos θ)L)$, where $l$ is the run length, $L$ the length of the channel and $θ$ the average reorientation angle during tumbles[55]. In our case, we estimated $b_{diff} \sim 0.007$, with higher bias indicating attraction towards a chemical in the source reservoir.

**Data availability.** All the relevant data are available from the authors on request.

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

## Acknowledgements

We thank Olga Besharova, Verena Suchanek and Aleksandra Kolodziejczyk for providing strains and plasmids; Dr Christophe Beloin for providing Ag43 antibodies; and Dr Ned S. Wingreen for discussions. This work was supported by grant 294761-MicRobE from the European Research Council, grant R01 GM082938 from the National Institutes of Health and grants SO 421/11-1 and SO 421/12-1 from the Deutsche Forschungsgemeinschaft.

## Author contributions

L.L., R.C. and V.S. designed the experiments; L.L. and R.C. performed the experiments. All authors analysed the data and wrote the manuscript.

## Additional information

**Competing financial interests:** The authors declare no competing financial interests.

DOI: 10.1038/ncomms13979    **OPEN**

# Corrigendum: Chemotaxis towards autoinducer 2 mediates autoaggregation in *Escherichia coli*

Leanid Laganenka, Remy Colin & Victor Sourjik

*Nature Communications* 7:12984 doi: 10.1038/ncomms12984 (2016); Published 30 Sep 2016; Updated 15 Dec 2016

Two previous studies describing the effects of autoinducer 2 (AI-2) on biofilm formation and chemotaxis in *Escherichia coli* were inadvertently omitted from the reference list of this Article. A reference to these studies along with ref. 27 should have been provided in the Introduction, as follows: 'It has been previously reported that, in *E. coli*, AI-2 can modulate biofilm formation (González-Barrios *et al.* 2006) and virulence-related phenotypes such as chemotaxis, swimming motility and attachment to host cells *in vitro* (Bansal *et al.* 2008, ref. 27).' These findings should have also been mentioned in the Abstract, where the physiological role of AI-2 in *E. coli* and other bacteria is referred to, and in the sections discussing motility, chemotaxis and biofilm formation in *E. coli*.

González Barrios, A. F. *et al.* Autoinducer 2 controls biofilm formation in *Escherichia coli* through a novel motility quorum-sensing regulator (MqsR, B3022). *J. Bacteriol.* **188,** 305-316 (2006).

Bansal, T., Jesudhasan, P., Pillai, S., Wood, T. K. & Jayaraman, A. Temporal regulation of enterohemorrhagic *Escherichia coli* virulence mediated by autoinducer-2. *Appl. Microbiol. Biotechnol.* **78,** 811-819 (2008).

