## [Peer Review File · Nature Communications]

Reviewers' comments:

Reviewer #1 (Remarks to the Author):

This work presents an analysis of Ag-43 and curli-mediated auto aggregation by *E. coli* in response to the AI-2 quorum sensing autoinducer. The authors find that auto-aggregation requires chemotaxis to AI-2 and that cell-cell interactions, in turn, increase AI-2 signaling (as measured by expression of an AI-2-responsive target gene) and resistance to oxidative stress. The authors also show that these same genes/signals have effects on *E. coli* biofilm formation.

The work is exciting, however, it is a bit unbalanced. The authors investigate many different traits, motility, chemotaxis, auto aggregation, AI-2 signaling, oxidative stress, biofilm formation...and they try to make connections among all of these phenotypes. Some of the analyses are solid and some are preliminary and not as convincing. Indeed, the work is mostly rigorous and highly quantitative in the analyses presented in Figures 1-4, however, the analyses in Figures 5 and 6 are qualitative. Given the state of the art (for biofilms especially) today, these qualitative pictures will not do. Because so many different traits are studied, the paper is diffuse, the explanations are short (due I'm certain to journal space constraints), but unfortunately, lacking in depth.

I want to make sure the editor and authors know that the work is exciting. My view is that it would be far better if the authors perhaps completely focused on auto aggregation and AI-2 signaling and added to the depth of what are the most exciting and novel aspects of the work. They could focus on stress and biofilms in a different manuscript. This way, they could write out their thoughts more clearly, tell us how they come to their interpretations, and provide more data to underpin their claims.

In the Discussion, the authors present a very compelling model. I like it, but they never refer back to their figures. I kept wondering what experiment/what evidence was tied to each claim made in the model. Perhaps they could use their model as a guide and perform experiments to prove each point?

I have a few suggestions for the authors and editor to consider.

The crux of the main argument is that chemotaxis and AI-2 signaling are required for auto aggregation. However, the authors never try to prove their case because the *luxS* mutant has a motility defect. The authors need to decouple these phenotypes/pathways. They need to use an apparatus/mechanism/microscopy technique that allows the WT and perhaps the *lsrC* mutant and even the *luxS* mutant to chemotact to AI-2 and then they need to quantify that AI-2 taxis enhances autoaggregation. I know that the *luxS* mutant has a motility defect, but it should matter if exogenous AI-2 is present or not. This kind of experiment, in which gradients of attractants are established and chemotaxis is quantified, is well within the PI's wheelhouse. Otherwise, the main results and connections between the systems are implied, but they are not proven.

The above experiment is especially important because in Figure 2d the authors show that exogenous addition of AI-2 represses autoaggregation, yet their main argument is that AI-2 signaling induces autoaggregation. These are conflicting ideas. They need to sort out what role chemotaxis has, what role AI-2 signaling has, and what their combined roles are on autoaggregation to convincingly prove their very important point.

They should also consider collecting cell free culture fluids and directly measuring AI-2. They infer that

AI-2 is present in one experiment from a fluorescent construct to the *lsr* operon. If they authors quantified AI-2 they could directly state the consequences of AI-2 signaling on autoaggregation.

If the analyses in the second half of the paper are to remain, they need to be performed with the same depth as the work presented in Figure 1-4. Figures 5 and 6 need to be made more clear. Specifically, different mutants are assessed in panels 5a and 5b, and then only a subset from both of those panels is studied in Figure 5c. How can the reader make conclusions when we do not have data from each of the three analyses for all of the mutants? The same is true for Figure 6. Only a subset of the mutants in 6a is assessed in Figure 6b.

With respect to Figures 5c and 6b, what is the takeaway? These data need to be quantified. The biofilm field has moved well past simple pictures. In Figure 6b all of the pictures look identical to me?

Reviewer #2 (Remarks to the Author):

The Sourjik group presents their insight on the role of AI-2 mediated chemotaxis during autoaggregation of *E. coli*. They show that this autoaggregation is mediated by antigen-43 or curli and require motility and enhanced by AI-2 mediated chemotaxis to be initiated. Finally, they show that aggregation protects the bacteria from oxidative stress and helps to initiation biofilms.

The paper builds on various previous observations:

- autoaggregation requires antigen-43 or curli
- Tsr was shown to mediate chemotactic response to AI-2
- aggregation enhances stress resistance
- Ag43 has an impact on biofilm formation
- curli affects autoaggregation under certain growth conditions

However, the current manuscript connects these previous observation with new insights and completes a story how AI-2 mediated chemotaxis impacts autoaggregation and all these previously observed phenomena. The manuscript is very well written and easy to read. While the paper presents some new observations, it also confirms previously described experiments. e.g. It was previously proposed that chemotaxis might promote aggregation, but it was experiments only showed negative impact of chemotaxis on aggregation of other species. This current manuscript clearly demonstrates the role of chemotaxis and AI-2 signalling of *E. coli* in this process.

The paragraph on "Aggregation enhances oxidative stress resistance and biofilm formation" mostly confirms previous observations and validates those with more diverse mutations, but highlights that autoaggregation properties are more important for these processes than chemotaxis.

page 8; the impact of AI-2 addition on reduction of aggregation should be explained - is this due to the lack of directed chemotaxis and movement towards already established aggregates? If this is the case, the single cell motility should be examined in the presence of homogeneous and directed AI-2 concentrations. Can AI-2 actually activate chemotaxis in planktonic cells using different assays (i.e. no autoaggregation)?

page 9; the lack of aggregate dispersal of the *cheY* strains is promiscuous, the authors propose that this is due to lack of movements toward the aggregate. However, this does not explain lack of dispersal in my opinion. Do all chemotaxis deficient strains lack dispersal? Time lapse images should be performed to clearly demonstrate whether dispersal is reduced in the chemotaxis mutants or simply, the growth of the aggregates is reduced.

Time lapse imaging on the autoaggregation properties of the wild type and its derivatives could greatly fine tune the observations and the proposed mechanisms.

Reviewer #3 (Remarks to the Author):

The manuscript by Lagananka, Colin and Sourjik provides a conclusive set of evidence for the role of chemotaxis and sensing of gradients of the ubiquitous quorum sensing molecule AI-2 in the initiation of autoaggregation in *Escherichia coli*. The effect of chemotaxis as a promoting agent for autoaggregation occurs regardless of the major adhesin implicated in maintenance of cell-cell contacts in aggregates (curli fibers or Ag53). The authors demonstrate that autoaggregation depends on sensing of AI-2 gradients by binding to the Tsr chemoreceptor, likely via the LsrB binding protein which was proposed to mediate AI-2 chemotaxis via interaction with Tsr in previous studies. The authors also provide experimental evidence for the role of this behaviour in promoting cell collisions at low cell densities, in further augmenting biofilm formation on abiotic surfaces and in enhancing the ability of cells to overcome oxidative stress challenges. Therefore, the work here unifies a series of unappreciated connected behaviours that are likely to be widespread.

The work presented here is broadly significant for the following reasons: 1. The authors provide experimental evidence that bacterial autoaggregation is a controlled behaviour mediated by chemotaxis, but not uptake or metabolism of a signaling molecule 2. They show that chemotaxis-mediated autoaggregation on a signaling cue (AI-2) secreted by the aggregated cells themselves, a behaviour that was previously thought to be restricted to social eukaryotes. 3. The authors provides a physiological role for AI-2 function. Given that many bacteria produce AI-2 and that a majority of bacteria are motile and chemotactic and able to transition between planktonic and sessile lifestyles, the results are likely relevant to many other microorganisms.

The manuscript is well written and described results obtained through a variety of experimental approaches, thereby leading to robust conclusions. The data are analyzed and presented very clearly and make use of appropriate statistics.

The interpretations are mostly clear with the exceptions of those pertaining to the role of AI-2 sensing in aggregate dispersals which were somewhat confusing. The authors interpret the data in Fig. 3d and Fig. 3e to mean that a loss of chemotaxis "towards" the aggregates, likely due to accumulation of AI-2 is primarily responsible for the apparent stability of the aggregates formed by the $\Delta*cheY*$ cells. Indeed such accumulation would dissipate the AI-2 gradient and/or AI-2 accumulation may cause loss of motility. While this behavior would explain the observations that aggregates remain of the same size, how does this demonstrates that it is required for aggregate dispersal?

Are cells motile even after 120 min?

If degradation of AI-2 is not required for aggregate dispersal, is there a different signal for dispersal, which would also require chemotaxis motility?

Are there any metabolic effects resulting from AI-2 accumulation that could trigger dispersal mediated through chemotaxis?

Supplementary Fig. 6: the $\Delta*LsrB*$ mutant seem to form more microcolonies that the $\Delta*cheY*$ mutant, is this a correct interpretation?

Point-by-point response to Reviewers' comments

Reviewer #1

We thank the Reviewer for finding our work exciting, and for providing helpful suggestions on improving the clarity of presentation and discussion of our results in the manuscript.

The work is exciting, however, it is a bit unbalanced. The authors investigate many different traits, motility, chemotaxis, auto aggregation, AI-2 signaling, oxidative stress, biofilm formation...and they try to make connections among all of these phenotypes. Some of the analyses are solid and some are preliminary and not as convincing. Indeed, the work is mostly rigorous and highly quantitative in the analyses presented in Figures 1-4, however, the analyses in Figures 5 and 6 are qualitative. Given the state of the art (for biofilms especially) today, these qualitative pictures will not do. Because so many different traits are studied, the paper is diffuse, the explanations are short (due I'm certain to journal space constraints), but unfortunately, lacking in depth.

Our work indeed describes not only the mechanism of autoaggregation but also its physiological consequences. We believe, however, that it was important to connect these different aspects of the collective behaviour that are typically studied separately, as also acknowledged by other reviewers. We followed Reviewer's comments and quantified the biofilm data presented in Figures 5 and 6. The results of this quantification, shown in new Supplementary Figures 10 and 12c, fully support our conclusions. To facilitate interpretation of our results in the context of our model of aggregation, we further added a graphical representation of the model (Supplementary Figure 13) and discuss it in greater detail, including references to the figures showing corresponding experimental data.

I want to make sure the editor and authors know that the work is exciting. My view is that it would be far better if the authors perhaps completely focused on auto aggregation and AI-2 signaling and added to the depth of what are the most exciting and novel aspects of the work. They could focus on stress and biofilms in a different manuscript. This way, they could write out their thoughts more clearly, tell us how they come to their interpretations, and provide more data to underpin their claims.

As mentioned above, we now present and discuss our model of autoaggregation in greater detail, and we also performed additional experiments that support it (see below). But although we agree that characterization of the AI-2-mediated autoaggregation is the most exciting and novel part of our work, we feel that demonstration of the physiological implications of autoaggregation is nevertheless very important and it adds further novel and general aspects to our study.

In the Discussion, the authors present a very compelling model. I like it, but they never refer back to their figures. I kept wondering what experiment/what evidence was tied to each claim made in the model. Perhaps they could use their model as a guide and perform experiments to prove each point?

As suggested by the Reviewer, we now added references to the key figures, to clearly refer to specific experiments while discussing our model. In addition, we made a graphical representation of the model (Supplementary Fig. 13).

The crux of the main argument is that chemotaxis and AI-2 signaling are required for auto aggregation. However, the authors never try to prove their case because the luxS mutant has a motility defect. The authors need to decouple these phenotypes/pathways. They need to use an apparatus/mechanism/microscopy technique that allows the WT and perhaps the lsrC mutant and even the luxS mutant to chemotact to AI-2 and then they need to quantify that AI-2 taxis enhances autoaggregation. I know that the luxS mutant has a motility defect, but it should matter if exogenous AI-2 is present or not. This kind of experiment, in which gradients of attractants are established and chemotaxis is quantified, is well within the PI's wheelhouse. Otherwise, the main results and connections between the systems are implied, but they are not proven.

We have now confirmed the chemotactic response of *E. coli* to the gradients of AI-2, as suggested by the Reviewer (new Supplementary Fig. 6), which is in agreement with a previous study cited in our manuscript (Hedge et al., 2011). We have further confirmed that this response requires LsrB. For the chemotaxis analysis, using $\Delta lsrB$ strain provides a more specific evidence of the AI-2 sensing than using $\Delta luxS$, since deletion of *lsrB* has no apparent effects on motility or chemotaxis besides the lack of the AI-2 response.

The above experiment is especially important because in Figure 2d the authors show that exogenous addition of AI-2 represses autoaggregation, yet their main argument is that AI-2 signaling induces autoaggregation. These are conflicting ideas. They need to sort out what role chemotaxis has, what role AI-2 signaling has, and what their combined roles are on autoaggregation to convincingly prove their very important point.

We apologize that this experiment was not sufficiently interpreted (this was also pointed out by other reviewers). There is no conflict between this experiment and our model; moreover, the observed repression perfectly supports the model because addition of high background of AI-2 impairs the ability of cells to sense gradients of AI-2 produced by the aggregates (as directly demonstrated in new Supplementary Fig. 6). As a result, chemotaxis towards aggregates and therefore the chemotaxis-mediated growth of the aggregates are suppressed. We now explain these observations in greater detail in the text.

They should also consider collecting cell free culture fluids and directly measuring AI-2. They infer that AI-2 is present in one experiment from a fluorescent construct to the lsr operon. If they authors quantified AI-2 they could directly state the consequences of AI-2 signaling on autoaggregation.

We believe that the transcriptional reporter used in our study provides reliable readout of the AI-2 levels in the cell-free culture fluids (Fig. 3e). We have now added a calibration of this reporter (Supplementary Fig. 8), which shows that its response is linear in the measured response range and allows us to quantify concentrations of AI-2 in the cultures during aggregation/disaggregation. We comment on this in more detail in the text.

If the analyses in the second half of the paper are to remain, they need to be performed with the same depth as the work presented in Figure 1-4. Figures 5 and 6 need to be made more clear. Specifically, different mutants are assessed in panels 5a and 5b, and then only a subset from both of those panels is studied in Figure 5c. How can the reader make conclusions when we do not have data from each of the three analyses for all of the mutants? The same is true for Figure 6. Only a subset of the mutants in 6a is assessed in Figure 6b.

We now quantified the structure (microcolony) formation within biofilms that are shown in Figure 5 and 6 (new Supplementary Figs. 10 and 12). We are also sorry for any possible confusion caused by panels *a* and *b* of Figure 5. They represent different experiments reporting two different physiological effects of autoaggregation – oxidative stress (panel *a*) and crystal violet biofilm staining (panel *b*) – and thus are not directly comparable to each other. To avoid confusion, we now rotated the graph in Figure 5a to make it look clearly distinct from Figure 5b. In panels Figure 5b and 5c that are indeed directly related, the same four strains are analysed.

In Figure 6, difference between panels *a* and *b* is purposeful. In Figure 6a, we show an overview of the effects of different mutations on autoaggregation. In Figure 6b, we only show strains that are most relevant to this study ($\Delta lsrB$, $\Delta cheY$ and Δflu) and where effects on the biofilm formation at 30°C have not been (well) characterized. Defects of $\Delta csgA$, $\Delta fliC$, $\Delta motA$ mutants in the biofilm formation have been already well described in previous studies, which are cited in the Discussion section, and we do not think it is necessary to repeat this characterization here.

With respect to Figures 5c and 6b, what is the takeaway? These data need to be quantified. The biofilm field has moved well past simple pictures. In Figure 6b all of the pictures look identical to me?

As mentioned above, we now quantified biofilm structure formation corresponding to Figure 5c and 6b has been quantified (Supplementary Figs. 10 and 12c); the biomass of the formed biofilm is also quantified (Fig. 5b and Supplementary Fig. 12b). This quantification fully supports our conclusions.

Reviewer #2

We thank the Reviewer for finding our work clear and conclusive and for acknowledging that it combines a number of previously known facts with new observations, and connects them into one coherent picture of AI-2 mediated autoaggregation. We also thank the Reviewer for pointing out several issues that require further clarification, which we address below:

page 8; the impact of AI-2 addition on reduction of aggregation should be explained - is this due to the lack of directed chemotaxis and movement towards already established aggregates? If this is the case, the single cell motility should be examined in the presence of homogeneous and directed AI-2 concentrations. Can AI-2 actually activate chemotaxis in planktonic cells using different assays (i.e. no autoaggregation)?

We apologize for not being sufficiently clear in interpreting this experiment (this was also pointed out by other reviewers). We now provide a much more detailed explanation in the text, along with additional experiments (new Supplementary Figs. 6 and 9). These additional experiments also confirm that AI-2 is a specific chemoattractant and does not affect motility or chemotaxis to other chemicals in planktonic cells.

page 9; the lack of aggregate dispersal of the cheY strains is promiscuous, the authors propose that this is due to lack of movements toward the aggregate. However, this does not explain lack of dispersal in my opinion. Do all chemotaxis deficient strains lack dispersal? Time lapse images should be performed to clearly demonstrate whether dispersal is reduced in the chemotaxis mutants or simply, the growth of the aggregates is reduced.

In Figure 3d, we added the data for $\Delta lsrB$, which also lacks chemotaxis towards AI-2 and, like $\Delta cheY$, does not disperse. Additionally, new Supplementary Fig. 9 shows that addition of high amounts exogenous DPD/AI-2, as well as L-serine or its non-metabolizable homologue Albu to pre-formed aggregates decreased aggregation rate and even induced disaggregation. On the contrary, $\Delta cheY$ and $\Delta lsrB$ strains remained insensitive to these chemoattractants, which supports the idea that excess of AI-2 leads to dispersal by inhibiting chemoattraction towards aggregates. An important point here (which was not sufficiently emphasized in the previous version of the manuscript) is that the Ag43-mediated cell association in aggregates is reversible, as can be clearly seen in the movies of cell aggregation. To stress this point, we have now added corresponding Supplementary Movies 1-4, as well as a Supplementary Figure 7. We argue in the text that this reversibility naturally leads to dispersion as soon as the chemotaxis-mediated self-attraction is suppressed.

Time lapse imaging on the autoaggregation properties of the wild type and its derivatives could greatly fine tune the observations and the proposed mechanisms.

Data shown in Figure 3a,b represent the analysis of the time lapse experiments. Additionally, we now included examples of movies corresponding to the early stages of aggregation of the wild-type and $\Delta cheY$ cells (Supplementary Movies 1 and 2), as well as movies showing dynamics of aggregates at the peak of aggregation (Supplementary Movie 3) and during dispersal (Supplementary Movie 4).

Reviewer #3

We thank the Reviewer for finding our work broadly significant and conclusive and for acknowledging that it connects a series of unappreciated connected behaviours. We also thank the Reviewer for pointing out that our interpretation of dispersal experiments needs to be presented in greater detail (as also noticed by the other reviewers).

The interpretations are mostly clear with the exceptions of those pertaining to the role of AI-2 sensing in aggregates dispersals which were somewhat confusing. The authors interpret the data in Fig. 3d and Fig. 3e to mean that a loss of chemotaxis "towards" the aggregates, likely due to accumulation of AI-2 is primarily responsible for the apparent stability of the aggregates formed by the $\Delta cheY$ cells. Indeed such accumulation would dissipate the AI-2 gradient and/or AI-2 accumulation may cause loss of motility. While this behavior would explain the observations that aggregates remain of the same size, how does this demonstrates that it is required for aggregate dispersal?

We now include Supplementary Movies 1-4 and Supplementary Figure 7, which show that aggregates formed via Ag43 interactions are highly dynamic, with cells leaving and joining aggregates. As a consequence, suppression of AI-2 chemotaxis towards aggregates would naturally lead to dispersal. This is now explained in the text more clearly. We also confirm that AI-2 has no apparent effect on motility or chemotaxis to other attractants (new Supplementary Fig. 6).

Are cells motile even after 120 min?

Yes, cells remain motile, as now shown in Supplementary Movie 4.

If degradation of AI-2 is not required for aggregate dispersal, is there a different signal for dispersal, which would also require chemotaxis motility?

No, as argued above we believe that suppression of AI-2 chemotaxis is sufficient to induce dispersal. Additional data for $\Delta lsrB$ strain now included in Figure 3d show that $\Delta lsrB$ cells lack dispersal similarly to $\Delta cheY$ cells, strongly suggesting that it is not simply chemotaxis, but specifically AI-2 chemotaxis that plays a role in dispersal.

Are there any metabolic effects resulting from AI-2 accumulation that could trigger dispersal mediated through chemotaxis?

As mentioned above, we do not see any evidence for indirect effects of AI-2 accumulation on motility or chemotaxis to other attractants (new Supplementary Fig. 6).

Supplementary Fig. 6: the $\Delta lsrB$ mutant seem to form more microcolonies that the $\Delta cheY$ mutant, is this a correct interpretation?

We have now performed quantification and statistical analysis of the microscopy data shown in Figures 5 and 6, which suggests that whereas $\Delta lsrB$ and $\Delta cheY$ strains are very different from the wild-type, they show no significant differences in 3D structure formation compared to each other (Supplementary Fig. 12c).

REVIEWERS' COMMENTS:

Reviewer #2 (Remarks to the Author):

The authors responded my comments adequately and strengthened the manuscript on requested points. I do not have further remarks.

Reviewer #3 (Remarks to the Author):

The authors have addressed all of my previous comments. The authors should also be commended for the additional information, including the supplemental movies, since these, together with the figures and modified text, greatly enhance the quality of the manuscript.

Manuscript NCOMMS-16-07856A

Point-by-point response to Reviewers' comments

No further issues were raised by the referees.